

# Identification and validation of PSMB7 as a novel biomarker for prognosis and immune infiltrates of lung adenocarcinoma

Yan Chen[1,*], Xin Ran[2,3,*], Ping Fu[1], Jie Ao[1], Guihua Zhu[1], Lianhua Zhao[1] and HuaLiang Xiao[1]

[1] Department of Pathology, Daping Hospital, Army Medical University, Chongqing, China
[2] Department of Cancer Center, Daping Hospital, Army Medical University, Chongqing, China
[3] Armed Police Hospital of Chongqing, Chongqing, China
[*] These authors contributed equally to this work.

Corresponding authors
Lianhua Zhao,
zhaolianhua206@tmmu.edu.cn
HuaLiang Xiao,
dpbl_xhl@tmmu.edu.cn

## ABSTRACT

**Background.** Lung adenocarcinoma (LUAD) is one of the most prevalent types of lung cancer globally; it is characterized by high incidence and mortality rates and contributes to over 1.8 million deaths annually. PSMB7, a crucial component of the 20S proteasome involved in protein degradation and antigen presentation, has been implicated in various cancers; however, its specific function in LUAD remains inadequately explored.
**Methods.** This research aimed to investigate the expression of PSMB7 in LUAD and its clinical significance using real-time quantitative PCR, immunohistochemistry, differential expression analysis, pathway enrichment analysis, immune cell infiltration, and DNA methylation.
**Results.** PSMB7 expression levels in LUAD tissues were considerably higher than those in the surrounding normal lung tissues and were associated with advanced pathological stages and poorer clinical outcomes. High PSMB7 expression was correlated with reduced overall and disease-specific survival. Functional enrichment analysis indicated that the differentially expressed genes associated with PSMB7 were mainly involved in protein–DNA complex assembly and chromatin remodeling. Moreover, LUAD tissues showed lower DNA methylation in PSMB7 promoters than that in normal lung tissues, which was correlated with reduced survival rates. A negative correlation was observed between PSMB7 levels and immune cell infiltration, particularly for effector memory T, B, follicular helper T, and mast cells.
**Conclusions.** We identified PSMB7 as a promising biomarker for LUAD prognosis because of its strong association with tumor progression and immune microenvironment modulation. Future studies should explore therapeutic strategies targeting PSMB7 to improve patient outcomes for LUAD.

## INTRODUCTION

Lung adenocarcinoma (LUAD) is the most common subtype of non-small cell lung cancer (NSCLC), accounting for approximately 40% of all lung cancer cases (*Kishikawa et al., 2022*; *Li et al., 2023*). LUAD ranks among the leading causes of cancer-related deaths globally, with approximately 1.8 million fatalities reported each year (*Ahn, Choi & Kim, 2021*). The incidence of LUAD has been increasing, particularly in nonsmokers and women (*Shan et al., 2013*). Despite some progress in diagnostic and treatment approaches, the prognosis for LUAD remains grim, with a 5-year survival rate <20% (*Tong et al., 2020*). LUAD is a multifaceted disease marked by a variety of characteristics and a wide range of molecular changes. This complexity suggests that tumors in different patients can have unique features, resulting in different responses to treatment (*Hua et al., 2020*). Although diagnostic and treatment methods have improved, many patients diagnosed with LUAD still have a bleak prognosis. This indicates an urgent need to develop and implement more effective diagnostic and treatment strategies.

The proteasome subunit beta type-7 (*PSMB7*) gene is a key component of the proteasome complex, which is crucial not only for protein degradation but also plays an important role in various cellular processes, such as cell cycle regulation, apoptosis, and DNA repair (*Ma et al., 2017*). Although the role of PSMB7 as a key subunit of the proteasome complex in various tumors has gradually gained attention (*Begum, Thota & Batra, 2023*), its function and mechanism of action in LUAD have not yet been systematically elucidated. Currently, a significant knowledge gap exists regarding the expression pattern of PSMB7 in LUAD, its clinical relevance, and relationship with tumor progression, and its specific role in regulating the tumor immune microenvironment. Despite sporadic reports suggesting that PSMB7 is associated with the occurrence and development of LUAD, its mechanisms of action in tumor cell immune evasion, immune cell infiltration, and regulation of DNA methylation remain unclear. Furthermore, existing studies lack a systematic validation of the applicability of PSMB7 as a prognostic biomarker and its potential value in immunotherapy (*Zhang et al., 2024*). Therefore, a thorough investigation of the expression characteristics of PSMB7 in LUAD, its clinical significance, and its interactions with the immune microenvironment, as well as clarification of its molecular mechanisms, will provide an important theoretical basis for revealing new mechanisms of LUAD development and identifying potential therapeutic targets.

In this study, we aimed to systematically assess the transcriptional and protein expression levels of PSMB7 in LUAD; analyze its association with the patients' clinicopathological features, prognosis, and tumor immune infiltration; and explore the impact that DNA methylation modifications have on expression and prognosis. Through these multidimensional, comprehensive analyses, this study aimed to fill the knowledge gap regarding PSMB7 in the field of LUAD, reveal its potential as a prognostic and immune-related molecular biomarker, and provide experimental evidence for subsequent therapeutic strategies targeting PSMB7.

## MATERIALS & METHODS

### Sample collection

From April to May 2025, anatomically matched tumor and adjacent normal tissues were collected from 15 treatment-naïve patients with LUAD who underwent surgical resection at the Thoracic Surgery Department of Daping Hospital in Chongqing, China. Individuals with an unclear pathological diagnosis and insufficient clinical information were excluded. The freshly excised specimen is immediately fixed in 10% neutral formalin. The consistency of the histopathological diagnosis was confirmed through a dual-blind pathological review performed by two independent, certified pathologists. The clinicopathological characteristics are shown in Table S1. The Ethics Committee of the Third Affiliated Hospital of the Army Medical University approved this study (no. 300, 2024), and all patients provided signed informed consent. The methods used in this study strictly adhered to the principles of the Declaration of Helsinki, as revised in 2013.

### Real-time fluorescent quantitative PCR detection

Total RNA was extracted from paraffin-embedded tissues using nucleic acid extraction reagents (FFPE DNA/RNA; Aide Biotechnology, Guangzhou, China), according to the manufacturer's protocol. The purity and concentration of RNA were measured using a NanoDrop spectrophotometer (Thermo Fisher Scientific, Waltham, MA, USA), with all samples having an $A_{260}/A_{280}$ ratio of >1.9. First-strand cDNA was synthesized from 150 ng total RNA in a 20-μL reaction volume using RT Master Mix for qPCR II (MedChemExpress, Monmouth Junction, NJ, USA). Gene-specific primers were designed using Primer-BLAST (NCBI) and synthesized by Sangon Biotech (Shanghai, China), followed by purification using polyacrylamide gel electrophoresis (>99% purity). The primers used were as follows: PSMB7 forward (5′-CAACTGAAGGGATGGTTGTTGC-3′) and PSMB7 reverse (5′-GCACCAATGTAACCTTGATACCT-3′); GAPDH forward (5′-CAGGAGGCATTGCTGA -TGAT-3′) and GAPHD reverse (5′-GAAGGCTGGGGCTCATTT-3′). Specificity: melt curve analysis confirmed single peak amplification (Tm: PSMB7 = 86.5 ± 0.3 °C; GAPDH = 87.1 ± 0.2 °C) with no primer dimer peaks (<80 °C). Efficiency: the standard curves generated from a 5-fold series of cDNA dilutions (five points) provided amplification efficiencies of 98.5% ($R^2 = 0.999$) for PSMB7 and 101.2% ($R^2 = 0.998$) for GAPDH (optimal range: 90–110%). The quantitative PCR (qPCR) amplification reactions were performed in a 20-μL volume, with samples run on the Applied Biosystems 7500 real-time PCR system (Thermo Fisher Scientific) under the following conditions: initial denaturation at 95 °C for 2 min, followed by 40 cycles of amplification, denaturation at 95 °C for 10 s, and annealing at 60 °C for 30 s. Melt curve analysis was performed at the end of the PCR cycles: 60–95 °C (increment: 0.5 °C/s). Gene expression was normalized to that of *GAPDH* (reference gene) and calculated using the $2^{-\Delta\Delta Ct}$ method. Technical reproducibility: the intragroup coefficient of variation (CV) of the $\Delta Ct$ values was <1% in three replicates. Biological reproducibility: three independent biological replicates were analyzed (intergroup CV of $\Delta\Delta Ct$ < 5%) (raw data in Table S2).

## Immunohistochemistry

Immunohistochemistry (IHC) staining was conducted on lung tissue sections that had been fixed in formalin and subsequently embedded in paraffin, as well as on paired normal adjacent tissue sections, with a thickness of 3 μm. The sections were stained with an anti-human PSMB7 antibody (1:2000; mouse; TA504368S; OriGene, Wuxi, China), following the manufacturer's instructions. Finally, the sections were counterstained with hematoxylin. IHC staining results were observed under an Olympus optical microscope (Olympus BX41; Olympus, Tokyo, Japan). Each LUAD sample was evaluated by using the H-score calculation method. The calculation formula used is as follows: (percentage of weak intensity cells ×1) + (percentage of moderate intensity cells ×2) + (percentage of strong intensity cells ×3). The numbers 0, 1, 2, and 3 represent the intensities of cell staining. The H-score value ranged from 0–3.

## Bioinformatics analysis

All bioinformatics analyses were carried out using the Xiantao Academic Network Server (https://www.xiantaozi.com/) based on R software (v4.2.1; *R Core Team, 2022*). The following key R packages were used: ggplot2 v3.4.4 (visualization), stats v4.2.1 (statistical tests), survival v3.3.1 (KM/Cox models), GSVA v1.46.0 (immune infiltration), and clusterProfiler v4.4.4 (functional enrichment). For input normalization, raw count data from The Cancer Genome Atlas (TCGA)/GTEx were converted to TPM values using the Toil pipeline (*Vivian et al., 2017*), followed by $\log_2$(value + 1) transformation. We also analyzed the protein expression levels of PSMB7 using the Human Protein Atlas (HPA) database (*Digre & Lindskog, 2021*). All raw data used in this study can be accessed at  https://www.jianguoyun.com/p/DQekKeUQ3fe8DRi0_oQGIAA.

## PSMB7 expression profile analysis

Analysis specific to pan-cancer and LUAD: the RNA-seq TPM data of PSMB7 was $\log_2$-transformed. We used the Wilcoxon rank-sum test to compare differences in expression between the tumor and normal tissues; R packages used: ggplot2 v3.4.4, stats v4.2.1, and car v3.1-0. A paired *t*-test was used to compare the paired samples of TCGA-LUAD tumor and adjacent normal tissues. Diagnostic receiver operating characteristics (ROC) curves were generated using pROC v1.18.0.

## Clinical relevance analysis

We used the Kruskal–Wallis test to determine how PSMB7 expression correlates with the clinical features of LUAD. The RNA-seq and clinical data were filtered to exclude normal samples and cases with missing clinical records.

## Survival analysis

The Kaplan–Meier curve (R packages used: survival v3.3.1 and survminer v0.4.9) was used to evaluate overall survival (OS), and the patients were risk-stratified based on the optimal cutoff value. After screening the variables through univariate Cox regression (threshold *p*-value < 0.1), multivariate Cox regression (rms v6.3-0) was applied to establish a prognostic model, and the results were visualized using a forest plot. The primary clinical

endpoint, complete response (CR), was strictly defined according to the RECIST 1.1 criteria (*Eisenhauer et al., 2009*) as the complete disappearance of all target lesions, with data sourced from the TCGA clinical annotations. CR status, which reflects treatment efficacy, was also included as a key indicator for survival prediction in solid tumors (*Schwartz et al., 2016*), and the other covariates (TNM staging and ECOG performance status) were extracted from electronic medical records through standardized processes. The expression level of *PSMB7* was log2(TPM + 1)-normalized and input as a continuous variable into the model, with the proportional hazards assumption verified through Schoenfeld residual tests. Additionally, subgroup survival analyses stratified by clinical variables were performed to further assess the robustness of prognostic factors.

### Differential analysis and functional enrichment

When focusing on PSMB7, a volcano plot was used to display differentially expressed genes (DEGs), with a gray dashed line marking the threshold (plotted using ggplot2). We selected DEGs based on the following criteria: |log2 FC|>0.5 and adjusted *p*-value (FDR) < 0.05 (DESeq2 v1.38.0). We identified key pathways using co-expression heatmaps (Spearman correlation) and Gene Ontology/Kyoto Encyclopedia of Genes and Genomes (GO/KEGG) enrichment analysis (clusterProfiler v4.4.4, GOplot v1.0.2) combined with *z*-value calculations. GSEA was used to display the enrichment patterns.

### Functional enrichment analysis

For the GO/KEGG analysis, the following clusterProfiler parameters were used: pAdjustMethod = "BH," *p* value Cutoff = 0.01, and *q* value Cutoff = 0.05. GSEA analysis: hallmark gene sets (MSigDB v7.5), parameters: min GS Size = 10, max GS Size = 500, and FDR < 0.25.

### Methylation and immune infiltration analysis

We used the UALCAN database to analyze the methylation level of the PSMB7 promoter region, and MethSurv was used for methylation-mRNA correlation analysis. The TCGA-LUAD HM450 methylation data were obtained using the UALCAN (https://ualcan.path.uab.edu/) and MethSurv (https://biit.cs.ut.ee/methsurv) databases. The ssGSEA algorithm (GSVA v1.46.0) was used to assess immune cell infiltration, and the association with PSMB7 expression was displayed using lollipop plots and heatmaps. Based on the ssGSEA algorithm (*Hänzelmann, Castelo & Guinney, 2013*), the GSVA parameters were set as follows: gene set, 24 immune cell markers from *Bindea et al. (2013)*. Core parameters: method = "ssgsea," kcdf = "Gaussian," and tau = 0.25 (rank normalization weighting factor). We calculated immune infiltration status using the 24 immune cell markers obtained from the Immunity article (*Bindea et al., 2013*), including activated dendritic cells (DCs), B cells, CD8 T cells, cytotoxic cells, DCs, eosinophils, immature DCs, macrophages, mast cells, neutrophils, natural killer (NK) CD56bright cells, NK CD56dim cells, NK cells, plasmacytoid DCs, T cells, T helper cells, T central memory cells, T effector memory (Tem) cells, T follicular helper (TFH) cells, T gamma delta cells, Th1 cells, Th17 cells, Th2 cells, and T regulatory cells.

## Validation of the nomogram

Nomograms are statistical prognostic models that present data using simple graphics. Nomograms are used to assess risk and prognosis based on specific patient characteristics or biomarkers. In a nomogram, each indicator of a sample corresponds to a predicted value on a specific axis, and the final total score can predict survival rates at 1-, 3-, and 5-years. A nomogram was constructed using the "*rms*" and "*survival*" packages in R, based on independent prognostic factors. Calibration plots were used to validate the nomogram's effectiveness, and the concordance index was then calculated.

## Statistical analysis

Statistical analyses of intergroup comparisons were performed using Wilcoxon and Kruskal–Wallis tests, as well as paired $t$-tests. Spearman correlation was used to assess co-expression relationships. A Cox regression model was used to analyze the survival correlation ($p < 0.05$, considered significant).

# RESULTS

## Elevated PSMB7 expression in LUAD

A comprehensive pan-cancer investigation indicated that the expression levels of PSMB7 were substantially increased in most tumor types, including bladder, breast, and colorectal cancers, compared with those in normal adjacent tissues (Fig. 1A). The concentration of PSMB7 in LUAD tissues was markedly elevated compared with that observed in normal tissues (Fig. 1B). In 58 LUAD tissues, the expression level of *PSMB7* was markedly elevated compared with that in the corresponding adjacent tissues (Fig. 1C). ROC curve analysis suggested that the expression level of PSMB7 could accurately predict LUAD, with an area under the curve of 0.712 (Fig. 1D).

Real-time qPCR (RT-qPCR) and IHC analyses were conducted to assess the expression levels of PSMB7 in LUAD tissues. Fifteen LUAD tissue samples and their adjacent normal tissues were randomly selected for RT-qPCR. The expression level of PSMB7 in LUAD tissues was substantially higher than that measured in the paired, adjacent, and normal tissues (Fig. 2A). The analytical data obtained from the HPA website (Figs. 2B, 2C) and IHC validation (Figs. 2D, 2E) indicated that the expression level of PSMB7 in LUAD samples far exceeded that in normal tissues.

## Association between PSMB7 expression and clinical pathological variables

The elevated expression of PSMB7 was strongly correlated with advanced pathological stages, especially when contrasting stage III with stage I (Fig. 3A), T4 with T1 (Fig. 3B), and N2 with N0 (Fig. 3C). Elevated expression levels of PSMB7 were strongly correlated with adverse clinical outcomes, particularly in the comparison between CR and progressive disease (Fig. 3D), as well as OS and disease-specific survival (DSS) (Figs. 3E, 3F).

## Prognostic value of PSMB7 in LUAD

The KM method was used to assess the relationship between PSMB7 levels and LUAD prognosis. To categorize patients according to PSMB7 expression, a minimum *p*-value
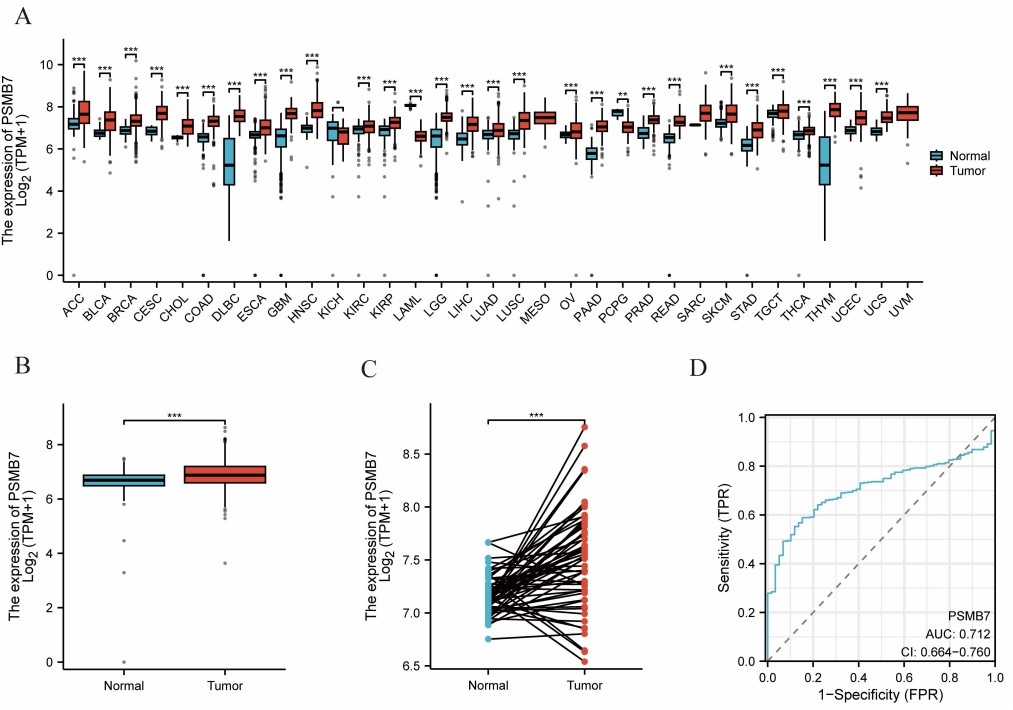

**Figure 1** **The expression level of PSMB7 in LUAD.** (A) PSMB7 is highly expressed in many solid tumors, including LUAD tissues (B, C). The ROC curve area was 0.712 (D), indicating that PSMB7 may serve as a diagnostic biomarker for LUAD. *P*-values were calculated using a two-tailed unpaired Student's *t*-test, with $*p < 0.05$, $**p < 0.01$, $***p < 0.001$. LUAD, lung adenocarcinoma; PSMB7, proteasome subunit beta type-7; ROC curve, Receiver Operating Characteristic curve.

was set as the threshold, and the patients were then divided into high- and low-expression cohorts. High PSMB7 expression correlated with a substantially poorer OS (Fig. 4A) and DSS (Fig. 4B). Furthermore, individuals exhibiting elevated PSMB7 expression experienced poorer prognoses at different stages of pathology (particularly between stages I and III and stages II and III), different tumor size categories (notably T1 *vs.* T4 and T1 *vs.* T3), the presence or absence of lymph node metastasis (N0 *vs.* N2), the presence or absence of distant metastasis (M0 *vs.* M1), clinical treatment results in progressive disease *versus* CR, individuals aged >65 years, and smokers (Fig. 5). Univariate and multivariate Cox regression analyses were performed to elucidate prognostic factors for patients with LUAD. In patients with LUAD, the expression of PSMB7, N2 stage, stable disease condition, and CR were identified as independent factors in determining OS (Fig. 4C).

## Functional enrichment analysis of DEGs associated with PSMB7

Differential analysis using DESeq2 identified 424 differentially expressed coding genes between the high and low PSMB7 expression groups, including 205 upregulated and 219 downregulated genes (adjusted *p*-value <0.05, |log2 FC|>0.5). Spearman correlation analysis of the top 10 most significant DEGs (*TSPAN6*, *TNMD*, *DPM1*, *etc.*) with PSMB7 expression showed a significant synergistic or negative correlation (Figs. 6A, 6B). The GO enrichment analysis results indicated that the aforementioned DEGs were

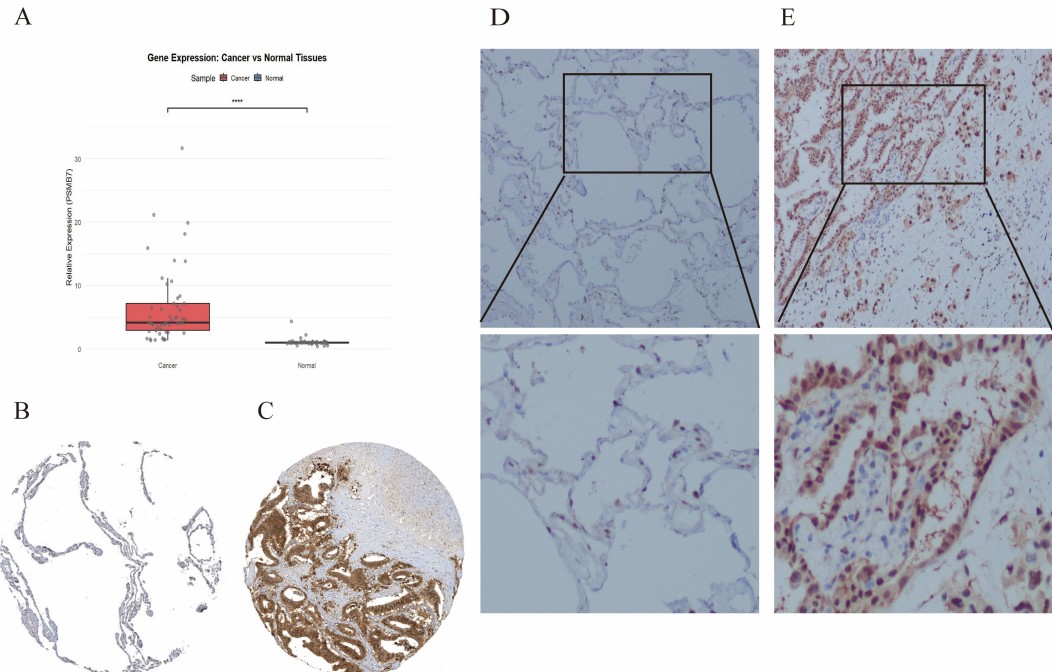

**Figure 2** **RT-qPCR and IHC were used to detect the level of PSMB7 in LUAD.** RT-qPCR (A) The results showed that the PSMB7 level in LUAD tissues was higher than that in adjacent tissues. (B and C) PSMB7 protein expression levels from the Human Protein Atlas database. (D and E) The differential expression of PSMB7 protein was verified by immunohistochemistry. LUAD, lung adenocarcinoma; IHC, immuno-histochemistry; RT-qPCR, Reverse Transcription Quantitative Polymerase Chain Reaction; HPA, Human Protein Atlas. Representative RT-qPCR and IHC results in 15 samples. (Comment: expanded by Zhu Guihua).

significantly enriched in biological processes related to protein–DNA complex assembly, chromatin assembly, and chromatin remodeling; cellular components mainly involved nucleosomes and protein–DNA complexes; and molecular functions included protein heterodimerization and transcriptional repression activity. KEGG pathway analysis revealed that the differential genes were primarily enriched in pathways related to olfactory transduction, systemic lupus erythematosus, and neutrophil extracellular trap formation (Fig. 6C; Table S3). The GSEA analysis results further suggested that the high PSMB7 expression group was significantly enriched in biological processes, such as the DNA double-strand break response (Fig. 6D).

## Relationship between PSMB7 expression and methylation status

The methylation analysis results from the UALCAN database indicate that the methylation level of the PSMB7 promoter region in LUAD tissues were significantly lower than those in normal pulmonary tissues ($p < 0.001$; Fig. 7A). Further survival analysis revealed that the OS rate of patients with LUAD in the low methylation group of PSMB7 was significantly lower than that in the high methylation group (Figs. 7B, 7C), suggesting that methylation modifications play an important role in the regulation of PSMB7 expression and prognosis of LUAD.

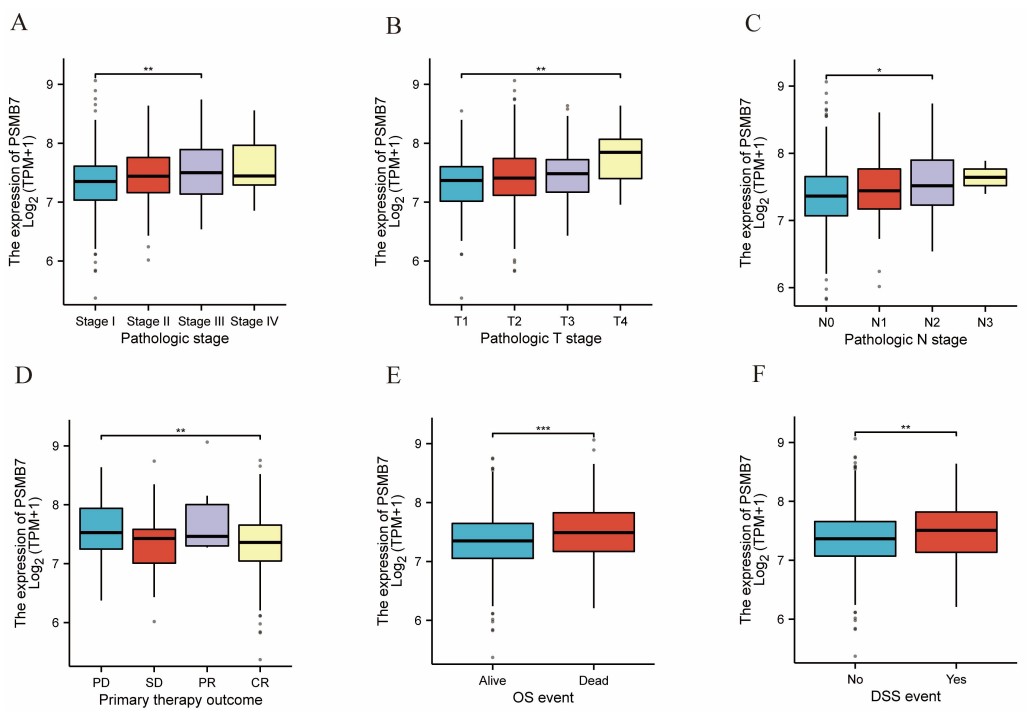

**Figure 3 Associations between PSMB7 expression and clinicopathological characteristics in LUAD.** Data are shown for (A and B) pathological stage, (C) N stage, (D) primary therapy outcome, (E) OS event, and (F) DSS event. $P$-values were calculated using a two-tailed unpaired Student's $t$-test, with $^{*}p < 0.05$, $^{**}p < 0.01$, and $^{***}p < 0.001$. CR, complete response; LUAD, lung adenocarcinoma; SD, stable disease; OS, Overall Survival; DSS, Disease-Specific Survival.

## Relationship between PSMB7 and immune infiltration

Immune cell infiltration analysis showed that the expression level of PSMB7 had a significantly negative correlation with the infiltration of certain immune cells, especially Tem, B, TFH, and mast cells (Fig. 8A). The ssGSEA results indicated that enrichment scores of the aforementioned immune cells in the high PSMB7 expression group were significantly lower than those in the low expression group (Figs. 8B–8I). Additionally, the expression level of PSMB7 was positively correlated with PD-L1 (CD274) ($p < 0.05$; Fig. S1), suggesting that it may be involved in regulating the immune microenvironment in LUAD.

## Development and validation of a nomogram based on independent variables

A nomogram incorporating various independent variables was created to estimate the clinical outcomes of patients diagnosed with LUAD. As the total score on the scale increased, the adverse factors worsened, indicating a poorer prognosis for patients with LUAD (Fig. 9A). Calibration curves were used to evaluate the precision and dependability of the predictive capabilities of the nomogram (Figs. 9B–9D). The results demonstrated that PSMB7 is an effective and independent prognostic marker of the outcome in individuals diagnosed with LUAD.

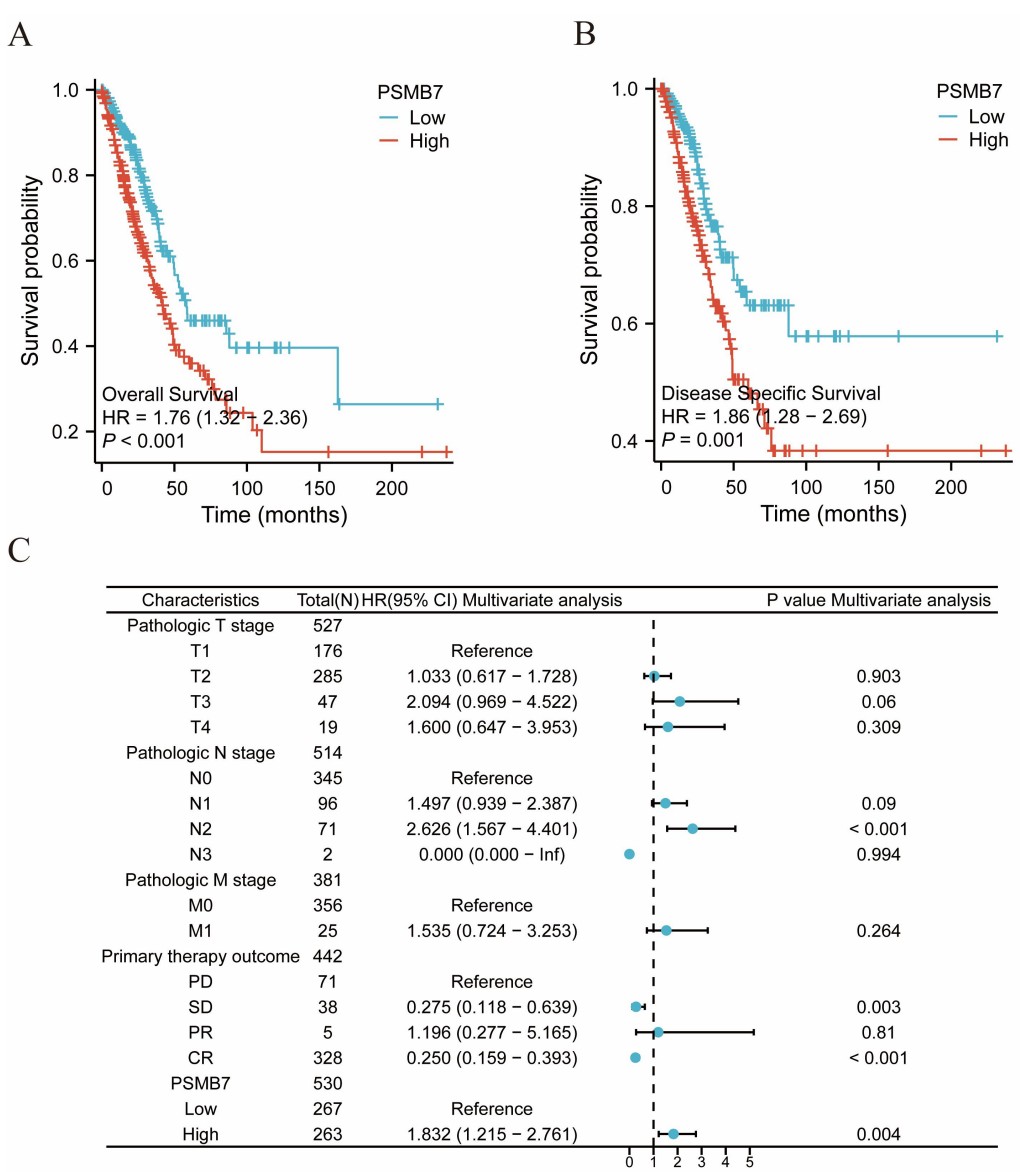

**Figure 4  The impact of PSMB7 levels on prognosis in lung adenocarcinoma patients was evaluated using Kaplan–Meier analysis.** (A) OS and (B) DSS in patients with lung adenocarcinoma with high and low PSMB7 expression levels. (C) Forest plot depicting OS outcomes in patients with lung adenocarcinoma based on multivariate Cox regression analysis. DSS, disease-specific survival; OS, overall survival.

## DISCUSSION

In recent years, the diagnosis and treatment of LUAD have heavily relied on the discovery and application of biomarkers. Traditional driver genes, such as *EGFR* and *ALK*, play a core role in the molecular classification and targeted therapy of LUAD, particularly as EGFR mutations and ALK rearrangements guide the clinical application of targeted drugs, including tyrosine kinase inhibitors, achieving significant progress in improving the prognosis of some patients. However, these molecular markers primarily target specific

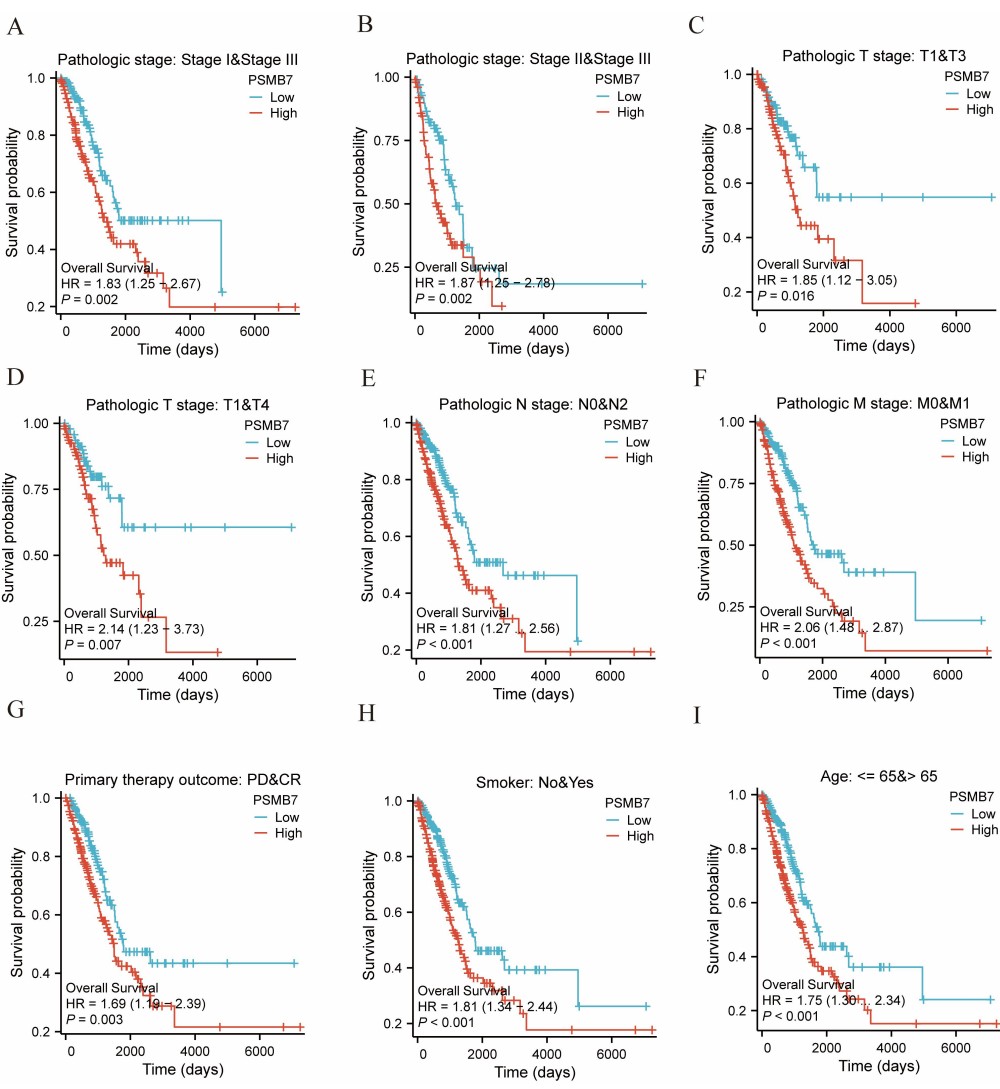

**Figure 5** **The impact of PSMB7 levels on prognosis across different subgroups of patients with LUAD as assessed by Kaplan–Meier analysis.** (A–I) OS curves of stage I and III , stage II and III , T1 and T3, T1 and T4, N0 and N2, M0 and M1, PD and CR, smoker, age, between high- and low- PSMB7 patients with LUAD. LUAD, lung adenocarcinoma; OS, overall survival; PD, Progressive Disease; CR, Complete Response.

molecular subgroups and have a limited impact on immune regulation and the tumor microenvironment. Additionally, Ki67, a classic marker related to cell proliferation, is often used to assess tumor growth rate and assist in prognosis judgment, but its specificity and independence are often limited in multivariate analyses.

In contrast, PSMB7, a proteasome subunit-related molecule, plays an important role in tumor protein degradation, antigen presentation, and immune regulation. The present study found that high PSMB7 expression in LUAD tissues was closely related to advanced pathological staging and independently predicted poor patient survival outcomes. Notably, the expression level of PSMB7 had a significantly negative correlation with tumor immune

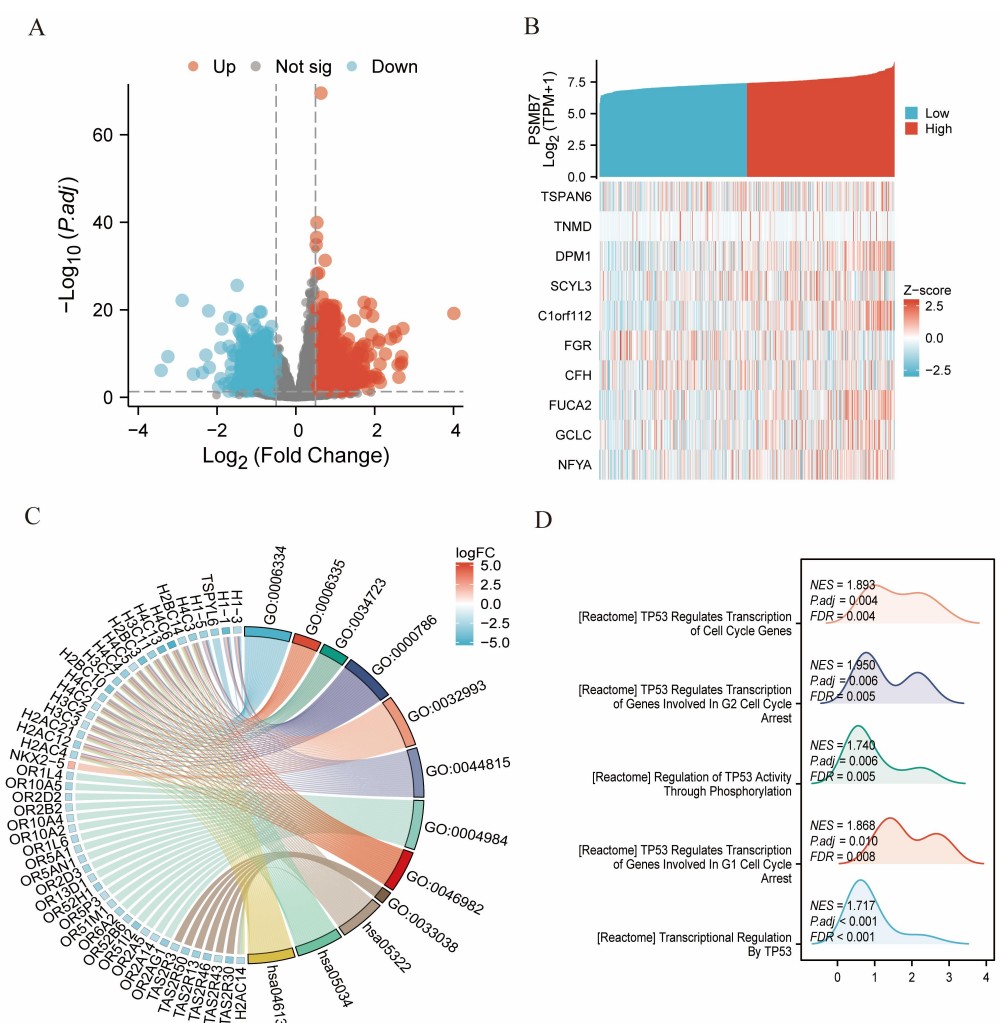

**Figure 6** **DEGs related to PSMB7 and its functional enrichment analysis utilizing GSEA, GO, and KEGG.** (A) Blue and red dots indicate significantly downregulated and upregulated DEGs in the volcano plot, respectively. (B) The top 10 DEGs were positively correlated with PSMB7 levels. (C) KEGG, GO, and GSEA. (D) Analyses of DEGs. DEG, differentially expressed gene; GO, Gene Ontology; GSEA, gene set enrichment analysis; KEGG, Kyoto Encyclopedia of Genes and Genomes. DEGs, Differentially Expressed Genes.

cell infiltration, particularly in Tem, B, TFH, and mast cells, suggesting its unique value in regulating the tumor immune microenvironment. Compared to molecular markers such as EGFR and ALK, the clinical application prospects of PSMB7 are more focused on prognostic assessment and the prediction of responses to immunotherapy, with broader applicability to populations and potential significance as a combined target for immunotherapy (*Zhang et al., 2024*). Furthermore, compared to proliferation markers such as Ki67, PSMB7 not only reflects the biological behavior of tumors but also participates in regulating tumor-associated immune mechanisms, allowing for a complementary role in prognosis assessment and clinical decision-making.

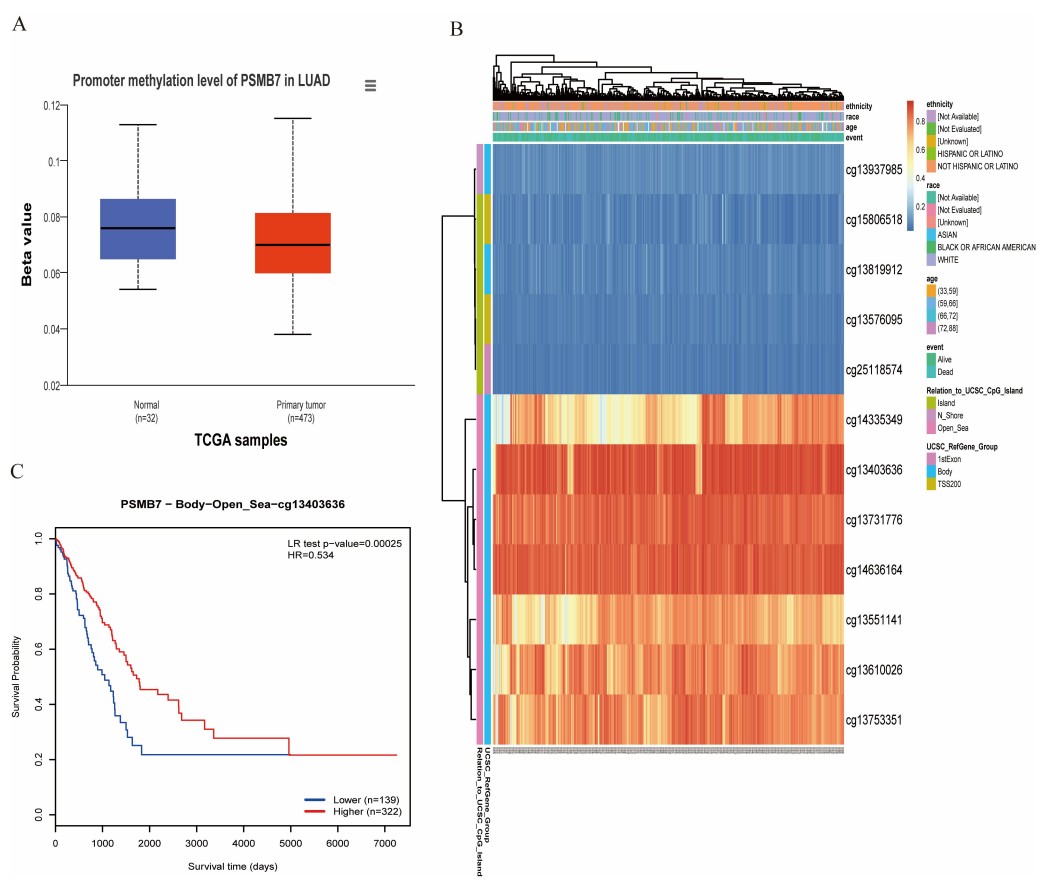

**Figure 7** **DNA promoter methylation levels of PSMB7 and its impact on prognosis in patients with LUAD.** (A) Promoter methylation level of PSMB7 in LUAD was significantly lower than that in normal lung tissue. (B) The correlation between *PSMB7* mRNA expression and promoter methylation. (C) Kaplan–Meier survival curves for different PSMB7 methylation sites. *P*-values were calculated using a two-tailed unpaired Student's *t*-test. LUAD, lung adenocarcinoma.

Therefore, PSMB7 is expected to serve as a novel molecular marker for prognostic assessment and the personalized treatment of patients with LUAD, especially in the era of immunotherapy, where its role in the regulation of the tumor microenvironment and immune evasion mechanisms warrants further in-depth study. Future multicenter, large-sample, mechanistic studies would help clarify the combined application value of PSMB7 with existing biomarkers, providing a more solid theoretical foundation and practical basis for the precise diagnosis and treatment of LUAD.

Our findings indicated that high PSMB7 expression was strongly associated with several negative clinicopathological factors, including advanced tumor stage, lymph node metastasis, and reduced OS. Moreover, an elevated PSMB7 level is an independent indicator of poor prognosis in patients with LUAD. These results suggest that PSMB7 plays a role in LUAD progression (*Zhang et al., 2024*). One potential reason for the link between high PSMB7 expression and unfavorable prognosis in patients with LUAD is its involvement in various biological processes through its catalytic function, which includes the regulation of

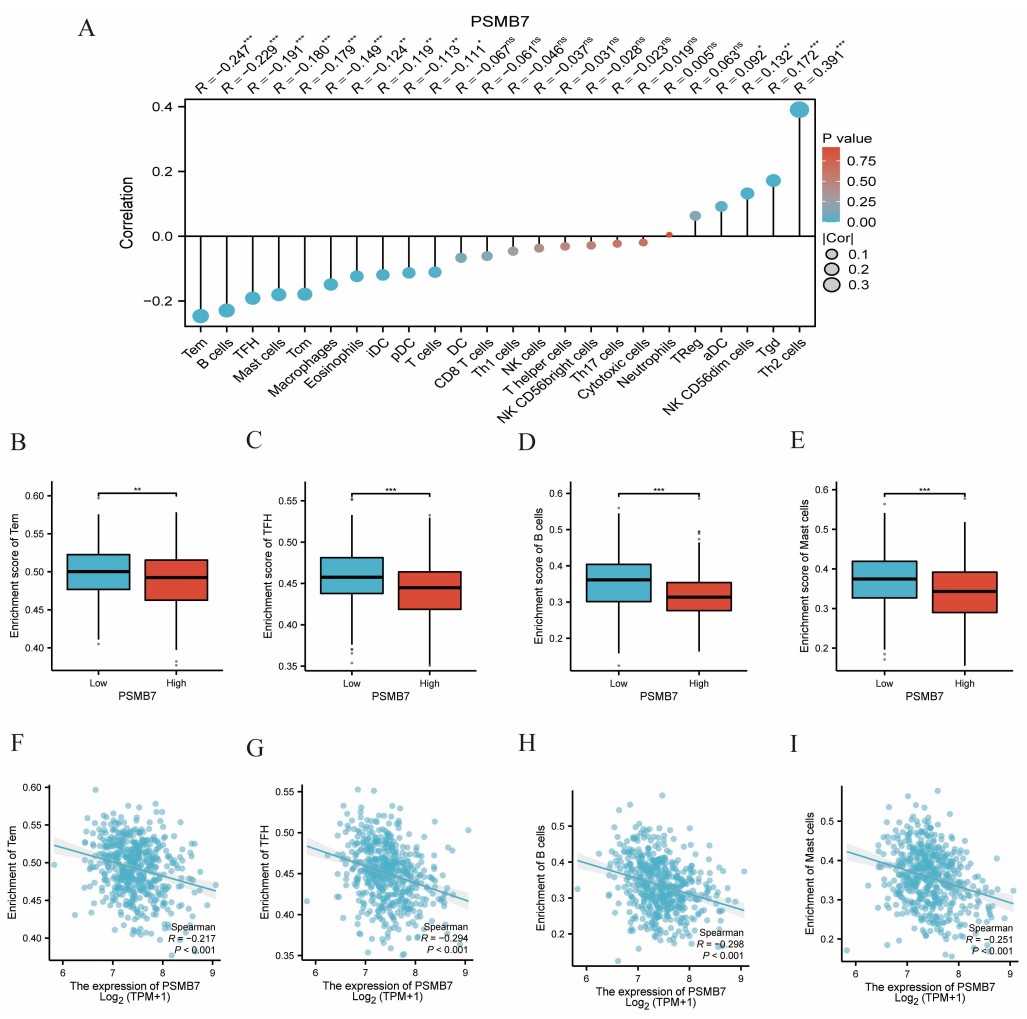

**Figure 8** **Correlation between PSMB7 level and immune cells infiltration in LUAD.** (A) Correlation between PSMB7 expression and 24 types of immune cells. (B–E) Comparison of immune infiltration levels of immune cells (including Tem, TFH, B cells and Mast cells) between high and low PSMB7 level groups. (F–I) The expression of PSMB7 was negatively correlated with the level of infiltrating immune cells, including Tem, TFH, B cells and Mast cells. *P*-values were calculated with two-tailed unpaired Student's *t*-test, * $p < 0.05$, ** $p < 0.01$, *** $p < 0.001$. LUAD, lung adenocarcinoma; Tem, effector T memory cell; TFH, follicular T helper cell.

the stability and degradation of intracellular proteins (*Huang et al., 2024*). This regulation can influence the proliferation and differentiation of tumor cells. The structural features of PSMB7 allow it to interact with other proteins, forming complex signaling networks that may contribute to cancer development.

Increased PSMB7 expression might aid tumor progression by promoting immune evasion within the tumor microenvironment and diminishing T cell-mediated immune responses (*Xing et al., 2022*). PSMB7 affects anti-tumor immune responses by hindering the effective presentation of tumor cell antigens, which may enable tumor cells to evade immune surveillance and proliferate. Tem cells are important components in mediating

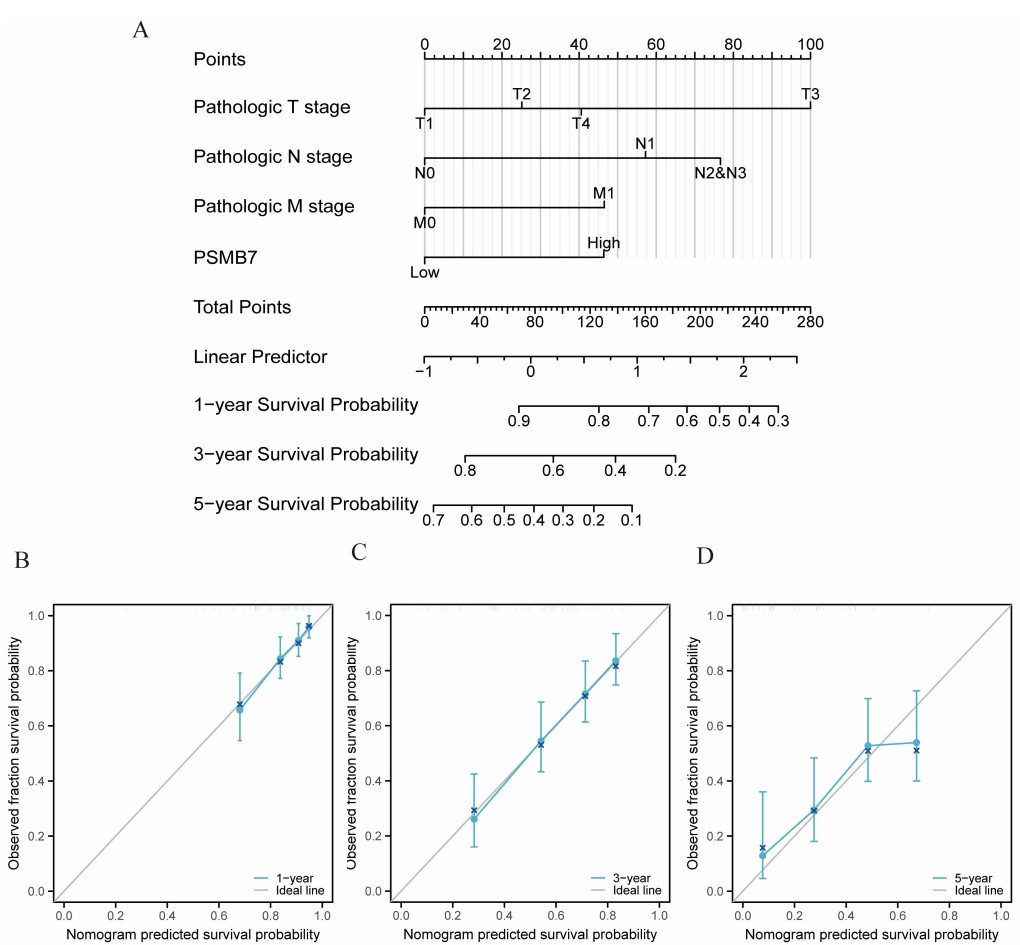

**Figure 9** **Calibration curves and nomograms for predicting OS rates in patients with LUAD.** (A) The nomogram chart provides a visual representation of the predicted OS rates for patients with LUAD at specific time intervals, such as one, three, and five years. (B–D) Calibration curves are graphical tools used to assess the accuracy of survival rate predictions at specific time points in patients with cancer. LUAD, lung adenocarcinoma; OS, overall survival.

anti-tumor immune responses, are capable of rapidly recognizing and killing tumor cells, and play a core role in tumor immune surveillance and immune therapy responses. The negative correlation observed between high PSMB7 expression and the level of Tem cell infiltration suggests that PSMB7 may weaken local anti-tumor immune effects by affecting the recruitment, survival, or function of Tem cells in the tumor microenvironment, thereby promoting immune evasion and tumor progression. Further investigation into the specific molecular mechanisms by which PSMB7 regulates Tem cell infiltration and function is needed, such as through *in vitro* cell experiments and animal models (*Yan et al., 2025*). Furthermore, high PSMB7 levels may promote the M2 polarization of tumor-associated macrophages, which are typically linked to tumor growth and metastasis. Elevated PSMB7 expression correlated with increased PD-L1 expression, which potentially facilitates tumor immune escape by modulating immune responses in the tumor microenvironment (*Hong*

*et al., 2023*). Specifically, PSMB7 may influence the stability and expression of PD-L1 by altering intracellular protein degradation pathways (*Ma et al., 2021*). In the current study, the expression level of PSMB7 was positively correlated with that of PD-L1, indicating that PSMB7 may substantially contribute to the immune evasion observed in LUAD (*Zheng et al., 2023*). This suggests that increased PSMB7 levels are not only linked to the inherent characteristics of tumor cells but are also closely associated with alterations in the surrounding immune microenvironment. Such immune suppression could enhance the resistance of tumor cells to both chemotherapy and immunotherapy, complicating efforts to effectively target and eradicate these cells. This mechanism may also explain the correlation between elevated PSMB7 expression levels and adverse outcomes in patients with LUAD.

Abnormal methylation patterns are closely related to the occurrence and development of various cancers, including LUAD, which is a prevalent form of NSCLC characterized by a complex interplay of genetic and epigenetic factors (*Fu et al., 2024*). Methylation affects not only the growth, movement, and invasive properties of tumor cells but also contributes to modulating the immune response within the tumor microenvironment, ultimately influencing patient outcomes (*Luo et al., 2022*). In the present study, we observed that increased PSMB7 expression was associated with lower methylation levels of its DNA. Specifically, patients with LUAD with low PSMB7 methylation levels had a poorer prognosis than those with high methylation levels. The methylation level of PSMB7 in LUAD tissues was lower and correlated with adverse clinicopathological features of the tumor. Generally, a low methylation status corresponds with the upregulation of PSMB7 expression and has the potential to boost the susceptibility of tumor cells to drugs used against cancer, thereby affecting patient prognoses. The methylation status of the *PSMB7* gene holds promise as a potential biomarker for LUAD; by measuring its methylation levels, we may gain insights into the tumor malignancy and survival prospects of patients.

PSMB7 has different biological functions and therapeutic significance in various malignancies. In LUAD, high PSMB7 expression is significantly associated with advanced pathological staging, immune microenvironment remodeling, and poor prognosis. It drives immune evasion by reducing Tem cell infiltration, promoting PD-L1 expression, and modulating epigenetic regulation (hypomethylation), suggesting that it may serve as a potential predictive biomarker for immunotherapy responses. In contrast, in multiple myeloma, PSMB7 primarily mediates bortezomib resistance through the compensatory upregulation of proteasome activity (especially in synergy with PSMB5/PSMB6) and activates the NF-$\kappa$B pathway to maintain malignant plasma cell survival, with its dynamic expression characteristics closely associated with disease staging (*Wu et al., 2021*). In breast cancer, the pro-cancer role of PSMB7 is prominently reflected in the independent prognostic value of chemotherapeutic drug (anthracyclines/paclitaxel) resistance and shortened disease-free survival, with its functional characteristics manifested in the regulation of multidrug resistance pathways rather than direct immune modulation (*Munkácsy et al., 2010*). In terms of therapeutic prospects, PSMB7, as one of the key subunits of the proteasome, has potential value as an important drug target. Proteasome inhibitors (such as bortezomib) have been widely used in the treatment of diseases like

multiple myeloma, and future specific inhibitors targeting PSMB7 are expected to expand new treatment strategies for solid tumors such as lung adenocarcinoma. However, the application of proteasome inhibitors in solid tumors still faces numerous challenges, including drug sensitivity, resistance mechanisms, and side effects. Relevant research foundations and clinical translation efforts need to be continuously advanced to explore more precise and effective treatment methods.

This study has certain limitations. Although we have expanded the sample size of wet experiments to 15 groups of LUAD and paired adjacent tissues, further enhancing the reliability of the results, the overall sample size is still limited, which might impact the study's statistical power and how widely the conclusions can be applied. Additionally, the AUC value for PSMB7 from large sample data is 0.712, suggesting that its effectiveness as a single-molecule diagnostic tool has limitations. In our study, we further validated its expression characteristics by increasing the clinical sample size and supplementing wet experiments. The results showed that PSMB7 is continuously highly expressed in LUAD tissues, giving more support for its potential clinical use. Future research could look into combining various molecular markers to explore joint diagnostic models to enhance the sensitivity and specificity of clinical diagnosis. Meanwhile, this study mainly focused on the expression characteristics of PSMB7, its prognostic value, and its correlation with the immune microenvironment and methylation levels, without exploring how PSMB7 is involved in immune evasion or chromatin remodeling, which should be confirmed with *in vivo* and *in vitro* experiments. Furthermore, the prognostic nomogram model constructed in this study was just validated internally in the TCGA database and lacks further validation in external independent cohorts. In the future, it is necessary to combine multi-center clinical samples to refine this model and boost its clinical usefulness. So, even though the results of RT-qPCR and IHC experiments match up well with the results of bioinformatics analysis, future studies should expand the sample size to further enhance the reliability and persuasiveness of the results. Secondly, this study mainly focused on the relationship between PSMB7 expression and prognosis, missing detailed studies on the specific functions of PSMB7 in tumor development and its interactions with the tumor immune microenvironment.

Future research should focus on the following directions. First, further *in vivo* and *in vitro* functional experiments are needed to elucidate the specific molecular mechanisms of action of PSMB7 during LUAD development, immune regulation, and drug resistance. Second, integrating multi-omics data for the systematic analysis of PSMB7-related pathways and their interactions with other key molecules is recommended to enhance our overall understanding of the biological characteristics of LUAD. Third, promoting the standardization of PSMB7 detection technology, developing clinical detection methods with high sensitivity and specificity, and conducting prospective validation in multicenter large cohorts are necessary to assess the clinical application value of PSMB7 as a diagnostic, prognostic, and treatment response predictive biomarker. Additionally, future studies could involve constructing multidimensional predictive models that combine various biomarkers (including immune-related, metabolism-related, and traditional tumor markers) to improve the accuracy of LUAD diagnoses and prognostic assessments. Finally,

targeted therapies based on PSMB7 and combined immunotherapeutic strategies are worth further development, which would be aimed at providing more precise and personalized treatment options for patients with LUAD.

In summary, our findings indicate that high PSMB7 expression levels are associated with advanced pathological stages and poor clinical outcomes, and that PSMB7 is important in tumor progression, the immune microenvironment, and methylation status.

# CONCLUSIONS

In brief, our study indicates that PSMB7 is upregulated in LUAD, and that high PSMB7 expression levels can predict a poor prognosis, providing important insights into the prognosis and treatment of patients with LUAD.

# ACKNOWLEDGEMENTS

We extend our sincere gratitude to all those who participated in this research.

## Funding

This research was supported by the Chongqing Intelligent Biological Manufacturing Engineering Research Center (Open Project Code 08-4800) and the Medical Research Project of the Chongqing Science and Health Federation (Key Project #2022ZDXM027). The funders had no role in study design, data collection and analysis, decision to publish, or preparation of the manuscript.

## Grant Disclosures

The following grant information was disclosed by the authors:
Chongqing Intelligent Biological Manufacturing Engineering Research Center: Open Project Code 08-4800.
Medical Research Project of the Chongqing Science and Health Federation: # 2022ZDXM027.

## Competing Interests

The authors declare there are no competing interests.

## Author Contributions

- Yan Chen conceived and designed the experiments, performed the experiments, analyzed the data, prepared figures and/or tables, authored or reviewed drafts of the article, and approved the final draft.
- Xin Ran performed the experiments, analyzed the data, prepared figures and/or tables, and approved the final draft.
- Ping Fu performed the experiments, prepared figures and/or tables, and approved the final draft.

- Jie Ao performed the experiments, prepared figures and/or tables, and approved the final draft.
- Guihua Zhu performed the experiments, prepared figures and/or tables, and approved the final draft.
- Lianhua Zhao conceived and designed the experiments, analyzed the data, authored or reviewed drafts of the article, and approved the final draft.
- HuaLiang Xiao conceived and designed the experiments, authored or reviewed drafts of the article, and approved the final draft.

## Human Ethics

The following information was supplied relating to ethical approvals (i.e., approving body and any reference numbers):

The Ethics Committee of the Third Affiliated Hospital of Army Medical University approved the study (no. 300, 2024).

## Data Availability

Raw data is available in the Supplemental Files.

## Supplemental Information

Supplemental information for this article can be found online at http://dx.doi.org/10.7717/peerj.19958#supplemental-information.

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
