# Peer review of "Identification and validation of PSMB7 as a novel biomarker for prognosis and immune infiltrates of lung adenocarcinoma"

_PeerJ, doi:10.7717/peerj.19958_

## Round 0.1 · original submission · Major Revisions

Please address all the reviewers' comments.

Reviewer 1 ·

Basic reporting

The authors have submitted a research manuscript investigating the associations of PSMB7 in lung adenocarcinoma. A few points should be addressed before

- The manuscript is generally well-written in professional English, but there are some minor grammatical inconsistencies and awkward phrasings that could benefit from language editing.

- Literature is sufficient, however, bioinformatics analysis parts require further explanations in the current manuscript rather than mere citations to the literature.

- It is unclear what is meant by "researchers divided cancer patients obtained from TCGA into two main groups, 123 based on the minimum p-value associated with PSMB7 expression levels, labeled as the high 124 expression group and the low expression group". This kind of categorization usually involves expression level, such as the mean/median expression across the dataset

- Abbreviations should be clearly explained in the figure legends.

- The authors have referred to the Xiantao web server for most bioinformatics analyses. Although this database has been cited in the field, the reporting of the analysis parameters is not entirely clear. Please provide further details.

- The manuscript states that raw data have been provided. A clear statement on data availability and repository access should be added.

- In Figure 1c, it is unclear how the dots are connected between normal and tumor samples. Are these paired samplings from the same individuals? I find that unlikely considering the healthy tissue availability in TCGA. Please explain this graph in more detail.

- Consider adding statistical comparisons in Figure 7a.

- Please clearly state the source data of the analysis. Are these all originating from the same TCGA data, or are some extra patient samples considered here?

- Please describe patient selection criteria and possible demographic/histological data to make a stronger point about the wide applicability of the findings.

Experimental design

- A lot of the analyses were conducted using the Xiantao web server, which is not readily accessible. Please provide further details on how the analyses were conducted for reproducibility.

- What is indicated by the ROC analysis is not clear. It is said that "PSMB7 predicts LUAD". Is the comparison against the normal lung tissue, or the rest of the cancer types? If the latter is proposed, the expression range of PSMB7 doesn't seem to be selectively high in LUAD compared to other cancers. In either case, the justification needs to be better stated here.

- The statistical analyses are appropriate, but a justification for specific cutoffs (e.g., |log2FC| > 0.5 in differential expression analysis) would be beneficial.

Validity of the findings

- The conclusions are mostly well-supported by the results.

- A section on potential limitations and future research directions is included, but could be expanded.

·

Basic reporting

• Clarity and Language: The manuscript is written in clear, professional English. Minor grammatical errors are present but do not impede comprehension.
• Structure: Conforms to PeerJ standards. However, the introduction could better emphasize the specific knowledge gap in LUAD regarding PSMB7.
• Figures: Figures are relevant but require improved resolution/contrast in some of them, eg, ROC curve.
• Raw Data: Provided but lacks metadata descriptors (e.g., sample IDs in RT-qPCR files).

Experimental design

• Strengths:
o Paired LUAD and normal tissues enhance internal validity.
o Integration of bioinformatics (TCGA, GTEx) with experimental validation (RT-qPCR, IHC).
o Ethical approval and informed consent documentation are complete.
• Weaknesses:
o Small Sample Size: Only 6 LUAD cases for experimental validation, risking underpowered conclusions.
o Methodological Gaps:
 Primer validation (specificity, efficiency) and antibody details (clones, catalog numbers) are missing.
 No justification for selecting 24 immune cell types in infiltration analysis.
 Lack of mechanistic experiments (e.g., knockdown/overexpression to validate PSMB7’s functional role).

Validity of the findings

• Key Findings Supported:
o PSMB7 overexpression in LUAD tissues (Figs. 1-2).
o Correlation with advanced stage, poor survival, and reduced immune infiltration (Figs. 3-8).
o Inverse association between PSMB7 promoter methylation and survival (Fig. 7).
• Concerns:
o Diagnostic Utility: AUC of 0.712 (Fig. 1D) is moderate; clinical relevance requires validation in larger cohorts.
o Multivariate Analysis: Inclusion of variables like "complete response" (CR) in Cox regression (Fig. 4C) needs clarification (e.g., treatment context).
o Immune Infiltration: Biological significance of negative correlations (e.g., Tem cells) is not mechanistically explored.

Additional comments

• Strengths:
o Comprehensive multi-omics approach (mRNA, protein, methylation, immune contexture).
o Use of public databases (TCGA, HPA) strengthens bioinformatics rigor.
• Limitations:
o Sample Size: Experimental validation with n=6 limits generalizability.
o Mechanistic Insights: No functional studies to link PSMB7 to immune evasion or chromatin remodeling.
o Clinical Translation: Prognostic nomogram (Fig. 9) lacks external validation.

Recommendations for Revision
1. Expand Experimental Validation: Increase LUAD cohort size (n ≥ 30) to strengthen statistical power.
2. Methodological Details:
o Provide primer validation data (e.g., melt curves, efficiency) and antibody specifications.
o Clarify criteria for selecting immune cell types and pathways in enrichment analysis.
3. Mechanistic Studies: Include in vitro/vivo experiments (e.g., PSMB7 knockdown) to explore its role in immune modulation.
4. Improve Figures:
o Label axes clearly (e.g., "Log2(TPM+1)" in Fig. 1D).
5. Discussion Revisions:
o Address limitations (small sample size, lack of functional data).
o Compare PSMB7’s role in LUAD with other cancers (e.g., breast, myeloma).
o Discuss therapeutic implications (e.g., proteasome inhibitors targeting PSMB7).

Ethical and Data Integrity
• Ethics: Approved by the Third Affiliated Hospital of Army Medical University (No. 300, 2024). Consent forms are appropriately documented.
• MIQE Checklist: Partially complete. Missing desirable details (e.g., primer purification methods, inter-assay CV).
• Image Manipulation: No evidence of inappropriate manipulation detected.

Reviewer 3 ·

Basic reporting

1. Some of the figures are unclear and appear stretched, particularly Figure 2A, Figure 7A. The quality of the figures needs to be improved with high-resolution and vectorization.

2. The manuscript mentions a correlation with PD-L1 (Fig. S1), but the supplementary figures are not included.

3. Supplementary Table S1 is referenced but not provided in the manuscript, limiting access to enrichment data.

4. Some figures (e.g., Fig. 6C for GO/KEGG enrichment) are dense and difficult to interpret without detailed legends.

5. The manuscript contains minor grammatical errors and awkward phrasing (e.g., “LUAD organization” instead of “LUAD tissue,” “obviously higher” instead of “significantly higher”). These issues occasionally hinder readability.

Experimental design

1. The experimental validation relies on only six pairs of LUAD and normal tissues for RT-qPCR and IHC. This small sample size limits the generalizability of the findings and raises concerns about statistical power.

2. RT-qPCR and IHC Details: The methods section lacks specifics on controls (e.g., negative/positive controls for IHC) and normalization procedures for RT-qPCR beyond GAPDH. The choice of GAPDH as a reference gene is justified, but its stability in LUAD tissues should be validated.

3. The exclusion criteria for patient samples (e.g., “ambiguous pathological diagnoses”) are vague. Specify what constitutes ambiguity.

Validity of the findings

1. Bioinformatics Analyses: The criteria for selecting DEGs (|logFC| > 0.5 in methods vs. |log2FC| > 1 in results) are inconsistent, which could confuse readers. The rationale for focusing on the top 10 DEGs is unclear, and their relevance to LUAD is not well-explained.

2. The KEGG pathway analysis includes seemingly unrelated pathways (e.g., olfactory transduction, alcoholism), which are not contextualized for LUAD, reducing their relevance.

3. While the study establishes correlations between PSMB7 expression, prognosis, and immune infiltration, it does not explore the underlying mechanisms (e.g., how PSMB7 regulates tumor progression or immune evasion). The discussion mentions possible roles in protein degradation and PD-L1 stabilization but lacks experimental evidence.

4"Furthermore, high PSMB7 levels may encourage M2 polarization of tumor-associated macrophages, which are typically linked to tumor growth and metastasis." lacks experimental and research evidence.

Additional comments

1. The discussion effectively ties PSMB7 to LUAD prognosis and immune modulation but lacks comparison with existing biomarkers (e.g., EGFR, ALK, Ki67). How does PSMB7 compare in terms of specificity or clinical utility?

---

## Round 0.2 · accepted · Accept

Now that all of the issues raised by the reviewers have been addressed, the manuscript can be accepted for publication.

·

Basic reporting

-

Experimental design

-

Validity of the findings

-